

# Short communication: Updated CRN Denudation datasets in OCTOPUS v2.3

Alexandru T. Codilean[1] and Henry Munack[1]

[1]School of Science, University of Wollongong, NSW 2522, Australia

**Correspondence:** Alexandru T. Codilean (codilean@uow.edu.au)

**Abstract.** OCTOPUS v2.3 includes updated *CRN Denudation* datasets, adding 1,311 new river basins to the *CRN International* and *CRN Australia* collections. The updates bring the total number of basins with recalculated $^{10}$Be denudation rates to 5,611, and those with recalculated $^{26}$Al rates to 561. To improve data relevance and usability, redundant data fields have been removed, retaining only those relevant to each collection. Additional updates include the introduction of several new data fields: the
latitude of the basin centroid and the effective basin-averaged atmospheric pressure, both of which improve interoperability with online erosion rate calculators. Other new fields record the extent of present-day glaciers and their potential impact on denudation rates, as well as estimates of the percentage of quartz-bearing lithologies in each basin — providing a basis for evaluating data quality. The updated data collections can be accessed at https://octopusdata.org (last access: 01 Dec 2024). The *CRN International* and *CRN Australia* data collections can also be accessed via their respective digital object identifiers
(DOIs).

## 1 Introduction

The OCTOPUS database was released in 2018 and consisted of a global compilation of cosmogenic $^{10}$Be and $^{26}$Al measurements from modern fluvial sediment along with optically stimulated luminescence (OSL) and thermoluminescence (TL) data from fluvial sediment archives across Australia (Codilean et al., 2018). Hosted at the University of Wollongong, the database
was made accessible to the research community through an Open Geospatial Consortium (OGC)-compliant web service, ensuring standardised and seamless data sharing.

In 2022, a new version of the database (OCTOPUS v2) was released (Codilean et al., 2022), introducing significant backend improvements, an upgraded web application with enhanced functionality, and updates to existing data collections, alongside the addition of new datasets. The *CRN Denudation* collections were updated, and the fluvial luminescence data were expanded
to include OSL and TL ages from aeolian and lacustrine sedimentary archives, now forming the *SahulSed* collection (Codilean et al., 2022). The new version also introduced *SahulArch*, a compilation of OSL, TL, and radiocarbon ages from archaeological records in Sahul, namely, Australia, New Guinea, and the Aru Islands, connected by formerly lower sea levels (Saktura et al., 2023). Additionally, two *partner* collections were integrated: a global dataset of $^{10}$Be and $^{26}$Al exposure ages on glacial landforms (https://expage.github.io; last access: 01 Dec 2024), and a collection of late Quaternary non-human vertebrate fossil ages
from Sahul (Peters et al., 2019, 2021). Regarding system architecture, OCTOPUS v2 introduced substantial improvements by





migrating to Google Cloud Platform (GCP) with a modular design that fully leverages GCP's cloud services. Additionally, it adopted a fully relational PostgreSQL database, which organises data both hierarchically and thematically, enabling seamless integration of all constituent data collections. Non-cloud-native components like GeoServer and Tomcat were containerised with Docker, ensuring a fully reproducible platform on GCP. The new platform also maintained support for integration with desktop GIS applications through OGC standards, enabling direct access to geospatial data alongside the upgraded web application. Finally, version 2 of the OCTOPUS database introduced comprehensive online documentation, including detailed information about the relational database. Additionally, a GitHub repository was created to host both the documentation and source code, while supplementary materials not stored in the database were made available through a Zenodo community (see online documentation for details: https://octopus-db.github.io/documentation; last access: 01 Dec 2024).

Two additional updates in 2024 introduced new data collections: the *SahulChar* collection, containing sedimentary charcoal and black carbon records from Australia, New Guinea, and New Zealand (Rehn et al., 2024), and the *Indo-Pacific Pollen (IPPD)* collection, containing palaeoecological pollen data along with site and dating information from Australia and the Indo-Pacific region (Herbert et al., 2024) (both in v2.2). Additionally, *SahulSed* was expanded in v2.3 to include OSL and TL ages from coastal landforms. Versions 2.2 and 2.3 also introduced updated *CRN Denudation* datasets, adding 1,311 new river basins to the *CRN International* and *CRN Australia* collections. Key improvements to these datasets include the removal of redundant fields, preserving only those relevant to each collection, and the inclusion of new fields that improve interoperability with online erosion rate calculators (Balco et al., 2008; Marrero et al., 2016; Stübner et al., 2023), and enable evaluating the quality of recalculated denudation rates.

Since its launch in 2018, the OCTOPUS database has recorded nearly 2,400 unique data requests (Fig. 1), with approximately 80% of these requests focused on denudation rate data. The *CRN International* and *CRN Australia* collections have become important resources for the global geoscience community, facilitating synoptic studies at both regional (e.g., Sternai et al., 2019; Delunel et al., 2020; Chen et al., 2021; Codilean et al., 2021; Whipple et al., 2023) and global scales (e.g., Godard and Tucker, 2021; Zondervan et al., 2023; Halsted et al., 2024), as originally intended. Here we describe the updated *CRN International* and *CRN Australia* collections focusing on spatial data coverage, new data fields, and interoperability with online calculators.

## 2 The updated *CRN Denudation* datasets

As described in previous publications (Codilean et al., 2018, 2022) *CRN Denudation* consists of four collections. Two of these, namely *CRN International* and *CRN Australia*, are officially supported by the OCTOPUS project and include recalculated $^{10}$Be, and where available also $^{26}$Al denudation rates, and have been assigned digital object identifiers (DOIs). The remaining two collections — *CRN Large Basins* and *CRN Denudation UOW (in preparation)* — are included solely for completeness and available at https://octopusdata.org (last access: 01 Dec 2024). While they do not contain recalculated rates, *CRN Large Basins* does include published $^{10}$Be and $^{26}$Al denudation rate data where available, and both collections are updated with new studies in each release as these become available.





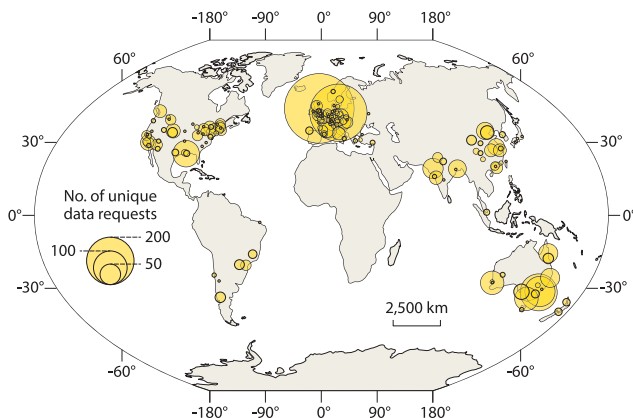

**Figure 1.** Geographic distribution of OCTOPUS data download requests recorded up to October 2024. Requests from outside Australia were predominantly for denudation rate data. Circle size represents the number of requests within each 25 km radius cluster.

The *CRN International* and *CRN Australia* collections comprise 5,611 $^{10}$Be and 561 $^{26}$Al recalculated basin-wide denudation rates, based on data published in the peer-reviewed literature up to the start of 2024. As part of the update to version

2.3, 1,311 $^{10}$Be and 163 $^{26}$Al values were added (see Fig. 2). We note that the extensive $^{26}$Al dataset ($n = 121$) from the recent publication by Halsted et al. (2024) has not yet been incorporated into OCTOPUS. Once included, this addition will significantly increase the number of $^{26}$Al data points available in the database. Collectively, the *CRN Denudation* collections encompass data from 290 studies. A recent literature search suggests that approximately 70 publications contain relevant data not yet captured in OCTOPUS. Therefore, version 2.3 of the OCTOPUS database currently includes about $80\%$ of all pub-

lished basin-wide denudation rate studies. Of the 70 missing studies, 47 contain fewer than ten $^{10}$Be basin-wide denudation rate estimates per study, with some featuring as few as two data points. Only 23 publications contain more than ten published data points per study, amounting to 363 additional $^{10}$Be denudation rates, and substantially fewer $^{26}$Al rates. Counting all 70 studies, version 2.3 of OCTOPUS captures roughly $90\%$ of published data points. However, given that the 47 studies with fewer than ten data points may not be added due to limited return on investment, the current version effectively captures $94\%$

of the viable data points for inclusion in the database.

In terms of topographic characteristics, the drainage basins of *CRN International* and *CRN Australia* span a wide range of mean elevations and slope gradients (Fig. 2a). The updated dataset has a geographical extent similar to previous versions (Codilean et al., 2018, 2022), with most data still sourced from Northern Hemisphere drainage basins (Fig. 2a). These basins primarily cluster in distinct, tectonically active regions, such as the Pacific coast of the United States, the Appalachians, the

European Alps, and the Tibet-Himalaya region (Fig. 2b). Coverage is also strong in the South American Cordillera. The version 2.3 update maintains this general geographical distribution (Fig. 2c), adding new data primarily from basins along the Alpine-Himalayan and circum-Pacific orogenic belts. Importantly, new basins also fill gaps in regions such as the Atlas Mountains in North Africa, the Caucasus Mountains, and the Tian Shan. However, data from low-gradient, tectonically passive regions





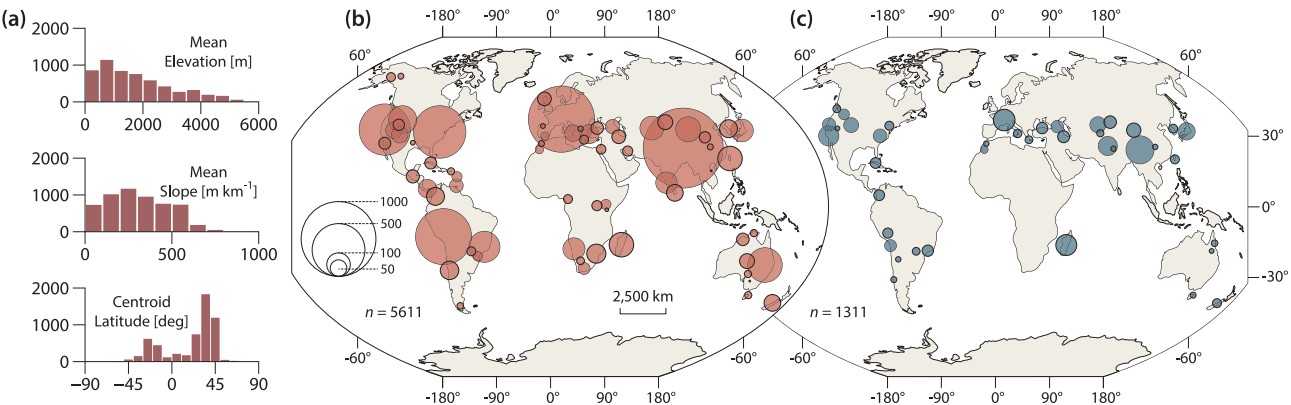

**Figure 2.** The updated *CRN International* and *CRN Australia* data collections. (a) The distribution of mean elevation, mean slope, and centroid latitude for the 5,611 basin included in version 2.3 of the OCTOPUS database. (b) The geographic distribution of the complete data collections and (c) that of the 1,311 newly added basins in version 2.3. Circle size represents the number of requests within each 250 km radius cluster.

remain sparse, particularly in Africa. Nonetheless, the update introduces a substantial dataset from Madagascar, as well as new
basins from southeastern Brazil and the Appalachians.

As with previous versions of OCTOPUS, for consistency across the *CRN Denudation* collections, published $^{10}$Be and $^{26}$Al concentrations ($\mathrm{atoms\,g^{-1}}$) were renormalised to the same AMS standards and basin-wide denudation rates ($\mathrm{mm\,kyr^{-1}}$) were recalculated with the open-source program CAIRN (Mudd et al., 2016). In addition to the vector and attribute data that is stored in the PostgreSQL + PostGIS database, the *CRN International* and *CRN Australia* collections also include seven raster
data layers and a series of text files representing CAIRN configuration and input / output files (see Codilean et al., 2022, for details). These additional files are organised in "studies" — each study represents one publication — and are saved as separated zip archives with names keyed to the unique STUDYID identifier assigned to each study. The provision of these data means that users can recalculate denudation rates using updated versions of the CAIRN code, keeping the CRN data reproducible and reusable into the future.

## 3  Interoperability with online calculators

CRN-based exposure ages and denudation rates require periodic recalculation as measurement standards and calculation protocols are regularly updated (Phillips et al., 2016; Schaefer et al., 2022). Online exposure age and erosion rate calculators (e.g., Balco et al., 2008; Marrero et al., 2016) play a crucial role in this process, offering platforms for dynamic and transparent data reduction. These tools harmonise disparate datasets and significantly enhance the reproducibility of cosmogenic nuclide
exposure ages and erosion rates (Balco, 2020). Although the CAIRN program operates independently of existing online calculators, it has been our preferred choice for several reasons. First, CAIRN is open-source and packaged in freely available software that runs on all commonly used operating systems. Second, it is automated and designed to support reproducibility;





users can publish a digital elevation model of their study area along with CRN data and CAIRN input files, ensuring denudation rates can be replicated. Third, CAIRN is part of LSDTopoTools, a suite developed for reproducible topographic analysis (Mudd et al., 2023), allowing CRN denudation rate calculations to seamlessly integrate with other topographic analyses within unified workflows. For computational efficiency, however, CAIRN only implements the time-independent Stone/Lal nuclide production scaling scheme (Stone, 2000) and so for applications where the use of more up-to-date scaling schemes is desired, users must turn to other packages (e.g., Charreau et al., 2019; Stübner et al., 2023) or make use of CAIRN's spatially averaged products for ingestion in available online calculators.

To facilitate interoperability with available online erosion rate calculators (e.g., Balco et al., 2008; Marrero et al., 2016), the *CRN International* and *CRN Australia* collections include two additional fields: CENTR_LAT, representing the latitude of the basin centroid, and ATM_PRESS, representing the effective basin-averaged atmospheric pressure. Both parameters are calculated by CAIRN. While the centroid latitude is straightforward, the atmospheric pressure is determined through an iterative process that identifies the pressure value best matching the spatially averaged Lal/Stone production rate (see also Mudd et al., 2016). Therefore ATM_PRESS should provide a fairly accurate single-value approximation of the spatially distributed altitude scaling factor for any given basin.

Using CENTR_LAT and ATM_PRESS with the Stone/Lal (St) scaling scheme in the Balco et al. (2008) online calculator yields $^{10}$Be denudation rates that very closely match those calculated with CAIRN (Fig. 3a). For example, the median difference between values calculated with the Balco et al. (2008) calculator and those calculated with CAIRN, when using the St scaling scheme is only $-0.6\%$ (mean = $-0.3\%$) with an interquartile range (IQR) between $-1.4$ and $0.4\%$, and total range between $-6.5$ and $6.8\%$. These values are bellow the analytical uncertainties of the measured $^{10}$Be concentrations recorded in the database, namely: $4.8\%$ (median), $7.4\%$ (mean), and $3.0 - 8.6\%$ (IQR); and well bellow the uncertainties of the calculated denudation rates. Recalculating denudation rates with the online calculator using the latitude of the basin outlet and the mean basin elevation instead of CENTR_LAT and ATM_PRESS, yields larger differences (Fig. 3b): $-3.1\%$ (median), $-4.5\%$ (mean), and $-5.6$ to $-1.6\%$ (IQR). Although these values are still within uncertainties of both $^{10}$Be concentrations and recalculated denudation rates, differences between the Balco et al. (2008) calculator and CAIRN correlate with basin relief (Fig. 3b) with the total range of differences being $-142$ to $23\%$. Therefore using the effective atmospheric pressure (ATM_PRESS) instead of the mean elevation is recommended as it reliably accounts for basin relief variations.

Basin relief covaries with basin area to some extent, with the largest basins in OCTOPUS also generally exhibiting the highest relief. To test how much of the scatter in Fig. 3b is due to basin relief versus basin area, we recalculated all denudation rates with the Balco et al. (2008) calculator using ATM_PRESS and swapping CENTR_LAT with the latitude of the basin outlet (Fig. 4). Results are virtually identical for basins smaller than $10^4$ km$^2$ and only become significant for basins larger than $10^5$ km$^2$ (Fig. 4b). Given that there are only 67 basins ($\sim 1\%$ of the total) in *CRN International* and *CRN Australia* that are larger than $10^5$ km$^2$, the choice of latitude will not matter for most applications. Nevertheless, we recommend using CENTR_LAT over the latitude of the basin outlet as the former is a better single-value approximation of the spatially distributed latitude scaling for any given basin.





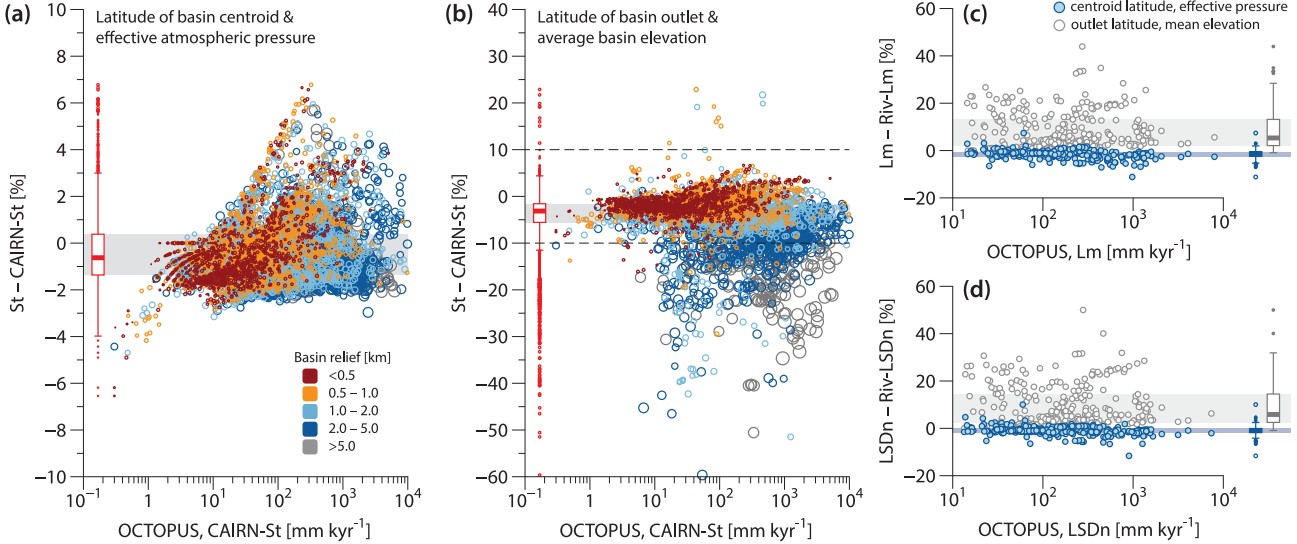

**Figure 3.** Comparison of ${}^{10}$Be denudation rates in the OCTOPUS database with those calculated using version 3 of the Balco et al. (2008) online erosion rate calculators. (a) Percent difference between ${}^{10}$Be denudation rates calculated using the St scaling scheme in the Balco et al. (2008) online calculators and those calculated with CAIRN (Mudd et al., 2016). Calculations are based on the basin centroid latitude and effective atmospheric pressure. (b) Same as (a), but using the latitude of the basin outlet and mean basin elevation. (c) Percent difference between ${}^{10}$Be denudation rates calculated using the Lm scaling scheme in the Balco et al. (2008) online calculators and those calculated with RIVERSAND (Stübner et al., 2023) using the same Lm scaling scheme. (d) Same as (c), but using the LSDn scaling scheme. In (a) and (b), the size and colour of symbols indicate the total basin relief (km). In (c) and (d), blue symbols represent values calculated using the basin centroid latitude and effective atmospheric pressure, while grey symbols represent values based on the latitude of the basin outlet and mean basin elevation. Shaded grey areas depict interquartile ranges (IQR), and the dashed horizontal lines in (b) denote the y-axis range from (a).

Since CAIRN only implements the time-independent St scaling scheme, we used RIVERSAND — a recently developed frontend to the Balco et al. (2008) calculators — to test the performance of `CENTR_LAT` and `ATM_PRESS` with time-dependent scaling schemes, specifically Lm (Nishiizumi et al., 1989) and LSDn (Lifton et al., 2014), as implemented in the Balco et al. (2008) calculator. RIVERSAND determines denudation rates based on river basin hypsometry (Stübner et al., 2023), allowing us to assess interoperability between OCTOPUS and online calculators. We selected eight studies from *CRN International* ($n = 290$ ${}^{10}$Be datapoints) that included high relief basins (median = 3.6 km; range between 0.6 and 8.7 km) and calculated denudation rates using `CENTR_LAT` and `ATM_PRESS` with the Balco et al. (2008) calculator and using the digital elevation model (DEM) provided via the OCTOPUS zip archives (see Sect. 2) with RIVERSAND. Differences between the two approaches are within a few percent (Fig. 3c and d) for both Lm (median = $-1.3\%$, mean = $-1.8\%$, IQR = $-2.5$ to $-0.5\%$) and LSDn (median = $-1.0\%$, mean = $-1.2\%$, IQR = $-1.8$ to $-0.1\%$) indicating that `CENTR_LAT` and `ATM_PRESS` are suitable approximations of basin characteristics even in high-relief settings.





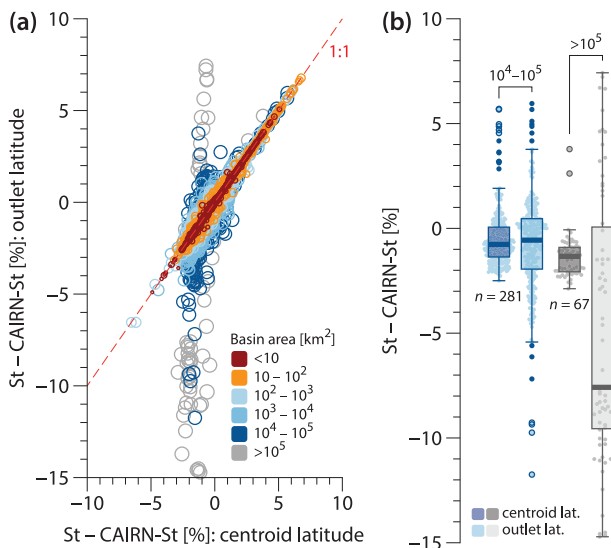

**Figure 4.** Comparison of $^{10}$Be denudation rates calculated using basin outlet latitude versus centroid latitude. (a) Percent difference between the St scaling scheme in Balco et al. (2008) and CAIRN, comparing results using outlet latitude versus centroid latitude. Symbol size and color represent the total basin area (km$^2$). (b) Box-plots showing the same as in (a) for basins with areas of $10^4 - 10^5$ km$^2$ (blue) and $> 10^5$ km$^2$ (grey).

## 4 Present-day glacier coverage and inferred denudation rates

According to the Global Land Ice Measurements from Space (GLIMS) database (Raup et al., 2007), 1,014 *CRN International*

basins are presently glaciated. While in just over $50\%$ of these basins ($n = 571$) present-day glacier coverage is below $5\%$ of the total basin area, the database includes 169 basins with present-day glacier coverage over $20\%$ and 16 basins where more than half of the basin area is covered by ice (Fig. 5). Not accounting for glacier-ice shielding will result in an overestimation of $^{10}$Be-derived denudation rates (Schaefer et al., 2022). To estimate the potential extent of this overestimation, we calculated end-member $^{10}$Be-derived denudation rates for the 1,014 affected *CRN International* basins using present-day glacier extents from

the GLIMS database. This calculation assumes that glaciated areas contribute sediment in proportion to their surface areas, with this sediment having a $^{10}$Be concentration of zero. While glaciated areas may indeed contribute sediment that is depleted in cosmogenic nuclides — due to both shielding from cosmic rays and potential excavation from depth — it is improbable that glaciated areas contribute sediment strictly in proportion to their surface areas. Therefore, our glacier-corrected $^{10}$Be denudation rates represent end-member minimum values, and the differences from the uncorrected $^{10}$Be denudation rates in

the OCTOPUS database reflect end-member maximum values.

To our knowledge, corrections for glacier coverage in denudation rate calculations are rarely attempted in the literature, primarily because doing so accurately is impossible due to lack of relevant data. Nevertheless, the median difference between corrected and uncorrected $^{10}$Be-derived denudation rates for basins with present-day ice occupying between $10-20\%$ of basin





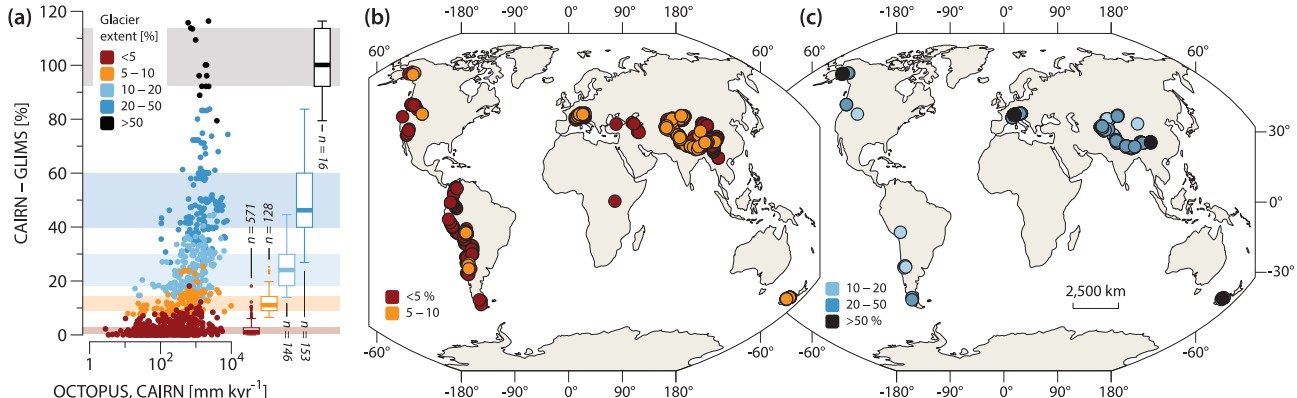

**Figure 5.** The influence of present-day glacier coverage on calculated $^{10}$Be denudation rates. (a) Percent difference between $^{10}$Be denudation rates calculated without glacier correction and those adjusted for the maximum impact of present-day glacier cover on diluting the $^{10}$Be-signal exported by rivers from each affected drainage basin (see text for details). Colours indicate present-day glacier extent (%), with box plots summarising percentage difference statistics for each colour band. Glacier extent data are sourced from the Global Land Ice Measurements from Space (GLIMS) database (Raup et al., 2007). (b) and (c) Geographic distribution of *CRN International* basins with present-day glacier coverage, presented on two maps for visual clarity.

area ($n = 146$) may be as high as $24\%$, and for basins with $20 - 50\%$ glacier coverage ($n = 153$), may be as high as $46\%$, for

example (Fig. 5), exceeding the uncertainties of the calculated denudation rates. While our calculations represent end-member values, they may still provide valuable insights into the potential influence of glaciers on the overestimation of $^{10}$Be-derived denudation rates when no correction is applied. To this end, the updated *CRN International* collection includes five additional data fields: `EBEGLA` and `EBEGLA_ERR`, which represent the glacier-corrected $^{10}$Be denudation rates and associated uncertainties as calculated by CAIRN; `EBE_DIFF`, indicating the percentage difference between $^{10}$Be denudation rates calculated

without glacier correction and those adjusted for maximum present-day glacier impact; and `GLA_KM2` and `GLA_PCNT`, representing the areal extent of present-day glaciers in $km^2$ and percentage, respectively. All data and calculations are based on present-day glacier extents from the GLIMS database.

## 5  Limitations

The primary goal of the OCTOPUS project was to create a comprehensive database of cosmogenic nuclide-based denudation

rates that is (1) globally consistent — achieved by recalculating all rates using a standardised methodology, and (2) reproducible and reusable — made possible by providing all input geospatial data, extensive metadata, and by using open-source tools, such as CAIRN. Adopting a global approach necessitates certain compromises. For instance, it requires selecting a DEM that is global or near global in extent and has a resolution suited to the range of basin areas included. Additionally, some corrections applied in individual studies — such as adjustments for quartz abundance variations or snow shielding — may be impractical

for a global database due to a lack of consistent global data for these specific corrections.



While the absence of detailed meteorological data on snow thickness prevents us from evaluating the potential bias introduced by omitting snow shielding corrections in our recalculated denudation rates, globally consistent, albeit low-resolution, lithological data are available. Although these data are insufficient for precise corrections related to quartz abundance, they may still be useful for assessing the overall quality of the recalculated data. To this end, *CRN International* and *CRN Australia*

include an additional field, labeled `QTZ_PCNT`, which indicates the percentage of basin area underlain by quartz-bearing rocks. This data was sourced from the Global Lithological Map (GLiM) by Hartmann and Moosdorf (2012). To reiterate, for most basins, these data are too coarse to enable precise corrections to the recalculated denudation rates; however, they may still offer valuable insights and help identify basins where such corrections could be warranted. For example, in only $\sim 65\%$ ($n = 3,681$) of the basins in *CRN International* and *CRN Australia*, quartz bearing rocks constitute more than $90\%$ of the basin area. Quartz

bearing rocks constitute less than half of the basin area in $\sim 15\%$ ($n = 869$) of basins, and less than $10\%$ of basin area in $\sim 6\%$ ($n = 351$) of basins.

To maintain consistency with previous database versions, the $^{10}$Be and $^{26}$Al denudation rates in the *CRN International* and *CRN Australia* collections were corrected for topographic shielding using the method outlined in Codilean (2006). However, a recent study by DiBiase (2018) suggests that topographic shielding corrections are generally unnecessary for calculating

basin-wide denudation rates, except in steep catchments where quartz distribution and / or denudation rates are not uniform. Nevertheless, the effect of topographic shielding corrections is minimal: omitting this correction in the basins with the $10\%$ highest topographic shielding values ($n \approx 410$) only shifts denudation rates by $3.5\%$ to $10\%$ — a range within the uncertainty of the calculated rates (see Codilean et al., 2022). As a result, while disregarding topographic shielding would yield slightly higher denudation rates, these differences would fall within the current uncertainty levels. Therefore, a recalculation of the *CRN*

*International* and *CRN Australia* collections without topographic shielding correction is not justified given the computational cost.

The CAIRN program requires input DEM data to be projected in one of the WGS84 UTM zones. This DEM is then used to derive a flow network, delineate individual basins, and link each grid cell within the basin to the sampling sites at the outlet. The requirement to project data into WGS84 UTM zones can pose challenges for large river basins that span multiple UTM

zones. In such cases, the DEM must be projected into one of these zones, which increases distortion in the resulting DEM. Moreover, the requirement to project data introduces challenges in low-relief regions — such as the interior of the Australian continent (Tooth, 1999) — where internally drained areas within drainage basins are common. Projecting the data in such areas can artificially connect these internally drained sub-basins to the main drainage network. (Fig. 6). Fortunately, due to the subdued topography in these regions, any errors in the calculated basin-wide production rates are minimal. Nevertheless,

basins with significant errors have been excluded from the database.

## 6 Conclusions

OCTOPUS v2.3 introduces updated *CRN Denudation* datasets, expanding the *CRN International* and *CRN Australia* collections with 1,311 additional river basins. These updates bring the total number of basins with recalculated $^{10}$Be denudation





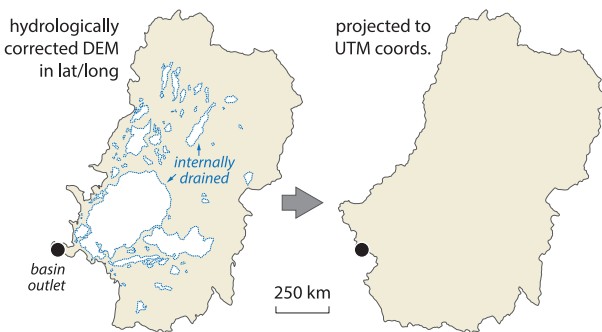

**Figure 6.** Drainage basin of the Murray-Darling river as derived from a hydrologically corrected DEM (left) versus that derived by CAIRN from the same DEM after projection to UTM coordinates (right). Note the absence of internally drained areas and the shifts in the location of the drainage divide following projection. See text for more details.

rates to 5,611 and those with recalculated $^{26}$Al rates to 561. Enhancements to data relevance and usability were achieved by re-

moving redundant fields, ensuring that only essential information remains for each collection. New data fields now include the latitude of each basin's centroid and its effective basin-averaged atmospheric pressure, improving interoperability with online erosion rate calculators. Additional fields track the extent of present-day glaciers and their potential influence on denudation rates, along with estimates of quartz-bearing lithologies in each basin, providing a framework for evaluating data quality. We hope that the OCTOPUS database will continue to ensure data reusability well beyond the scope of their initial collection,

thereby enabling large-scale, synoptic studies that would otherwise be unattainable.

*Data availability.* OCTOPUS v2.3 can be accessed at https://octopusdata.org (last access: 01 Dec 2024). Users should refer to the DOIs provided to ensure that they are accessing the current and supported version of the data. Comprehensive online documentation can be accessed at: https://octopus-db.github.io/documentation/ (last access: 01 Dec 2024). Supplementary data that was used in this paper but is not available via the OCTOPUS database, can be accessed at: https://doi.org/10.5281/zenodo.14014985 (Codilean and Munack, 2024).

*Author contributions.* ATC and HM contributed equally to the work presented here.

*Competing interests.* The contact author has declared that neither they nor their co-author have any competing interests.

*Acknowledgements.* We acknowledge financial support from the Australian Research Council Centre of Excellence for Australian Biodiversity and Heritage (ARC Grant CE170100015).





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
