# Peer review of "Short communication: Updated CRN Denudation collections in OCTOPUS v2.3"

_Geochronology, 2024_

## Author Comment (AC1)

*Preprint gchron-2024-28*
*Short communication: Updated CRN Denudation datasets in OCTOPUS v2.3*

We wish to thank the reviewer for taking time to consider our manuscript and for providing constructive criticism that will greatly improve the manuscript and ultimately the OCTOPUS effort.

We provide answers to each point below and hope that we can keep the discussion going and the reviewer will answer some of our queries before the public discussion phase closes on the 5th of January.

RC – Reviewer comment
AR – Author response

| RC1 | *My only major comment is that the authors should consider recalculating erosion rates using a time-variant scaling and removing topographic shielding, since they already acknowledge the problems with these approaches in the manuscript. The authors argue that this would cost too much computational time. However, all the basin-average effective atmospheric pressures have already been calculated already, therefore, the time-consuming pixel-based production rate calculation should be necessary anymore. Aren't all the necessary parameters for, e.g., Riversand now pre-calculated and the actual denudation rate calculation should be reasonably fast?* |
|---|---|
| AR1 | The answer to this comment is complicated:

There are a total of 284 studies in CRN International and CRN Australia and re-calculating denudation rates using CAIRN would take a minimum of 2 to 3 months with the IT resources that are available to us for this purpose. While recalculating some studies takes only a few minutes, others will take many hours, and a few will take days. One can distribute CAIRN processes across a cluster of computers, but we do not have access to this. Ignoring topographic shielding shifts the calculated denudation rate by a few percent. The 10% quoted by the reviewer (see RC6) is the difference obtained for the basin with the highest topographic shielding – so it is the absolute maximum – and in most case the difference will be a few percent. Compared to this, the median uncertainty on the calculated $^{10}$Be denudation rates is ~20%. Topographic shielding is a non-issue in our opinion, and probably not worth two to three months of recalculating time. Given that OCTOPUS provides all the data necessary for recalculating rates, however, means that these recalculations can be done on a case-by-case basis by the users themselves, if necessary.

Using RIVERSAND requires some processing of the CAIRN input data – such as reprojecting rasters and shapefiles (in CAIRN rasters are in UTM coordinates but sample data is in geographic coordinates; OCTOPUS exports shapefiles in Web Mercator coordinates) and fixing no-data issues (i.e., reclassifying no data pixels), and creating the required input table. Running RIVERSAND with the 284 studies will not take months but it is not a trivial exercise given the large number of studies. As we show in Figure 3, recalculating denudation rates using the basin centroid latitude and effective |

| | |
|---|---|
| | atmospheric pressure obtained from CAIRN produces values that are virtually identical to those obtained using RIVERSAND. Therefore, we do not see the advantage of using RIVERSAND in this case.

The basin-averaged effective atmospheric pressure is calculated for all CRN Int and CRN Aus basins and can be used with the Balco calculator – this is what was done for the purposes of Figure 3. However, for new studies added to the OCTOPUS database, we would still need to run CAIRN to calculate the basin-averaged effective atmospheric pressure before loading the values in the Balco calculator.

As we explain in Section 3, we prefer CAIRN over other approaches, mainly due to it being automated and also for it being part of LSDTopoTools. The latter allows CRN denudation rate calculations to seamlessly integrate with other topographic analyses within unified workflows – something not yet meaningfully exploited by people, but in our opinion something with great potential. Thus we are not keen on abandoning CAIRN. |
| RC2 | *Time-invariant scaling: As has been argued, e.g., by Greg Balco in a blog post https://cosmognosis.wordpress.com/2020/10/10/version-3-erosion-rate-calculator-benchmarked-finally/*

*time-invariance can really become a problem for slow erosion rates. The bias arises because the current magnetic field strength is high and was lower in the past, and most calibration data are from the past 20kyr, where field strength was high. I quote from the Balco blog: "Samples with lower erosion rates reflect production during longer-ago periods of weaker magnetic field strength and higher production rates, so an erosion rate computed with time-dependent scaling will be higher than one computed with non-time-dependent scaling. " Balco shows that this bias can be up to 40% and is therefore quite significant.*

*As the authors argue, many people download Octopus data for global studies and therefore use a large range of low and high erosion rates in their studies. In such a case, time-invariant production rates is a problem because it introduces a systematic bias. For instance, many studies investigate the non-linear relationship between erosion rate and river steepness (ksn) (Adams et al., 2020). Using time-invariant scaling and having a large range in erosion rates, the Ksn-E relationship would become more non-linear just due to the bias introduced by not accounting for magnetic field variation.* |
| AR2 | We fully agree with the reviewer and Greg's blog-post is what motivated us to improve interoperability between OCTOPUS and the Balco online calculators. |
| RC3 | *From my perspective, an option would be to switch from CAIRN to RIVERSAND (Stübner et al., 2023), as has been done for calculations in figure 3. I understand that requesting the recalculation of all rates using a time-dependent scaling scheme is a big ask, but I invite the authors to assess whether this is feasible.* |
| AR3 | See our response in AC1.

Given the similarities between the rates in Figure 3C&D, the question is not whether we should switch or not to RIVERSAND, but rather (1) whether we should include the |

| | |
|---|---|
| | recalculated Lm and LSDn rates in the database, or (2) whether we should – as we did in the current version – include the centroid latitude and effective atmospheric pressure in the data tables and let users run the recalculations themselves.

If we subscribe to the mantra of a transparent middle-layer for data management and analysis (as outlined in Balco 2020, 10.1016/j.quageo.2007.12.001), denudation rates are most reliable when freshly calculated. Therefore, providing the centroid latitude and effective atmospheric pressure in the data tables is the most appropriate approach – given that we tend to refresh OCTOPUS only every couple of years as a substantial number of publications become available.

We include input data for the Balco online calculators in the Zenodo repository created for out manuscript: https://zenodo.org/records/14014985. Should we also include this in the OCTOPUS data tables or should we also add the Balco calculator output to the OCTOPUS data tables?

**We would appreciate the opinion of the reviewer or of the broader community here.** |
| RC4 | *If the authors choose to stay with the CAIRN calculation, it would be valuable to show a comparison like in Fig. 3C/D, however, using the Octopus CAIRN-St rates versus the Riversand Lm or LSDn rates. The authors selected high-relief basins for their current approach. However, for a figure comparing the time-invariant and time-variant scaling schemes, the author should select studies that contain a large gradient in erosion rates.* |
| AR4 | We could certainly do this. However, both Greg Balco's blog-post and the RIVERSAND paper (https://doi.org/10.1017/rdc.2023.74) do an excellent job by showing such a figure (note that OCTOPUS CARIN-St rates are similar to the online calculator St rates for most basins – Figure 3B) and so we are not sure whether we would be contributing with anything new. |
| RC5 | *Lithology: More details are warranted for the estimation of quartz percentage in the basins. This is a really useful addition to Octopus. However, the authors do not describe, which lithology classes in GLiM are assumed to be quartz bearing. GLiM contains layers such as mixed sediment that can be full of quartz or devoid of it. Please, provide more detail on how this crucial number was estimated.* |
| AR5 | We agree fully. We did an embarrassing job here and we will add more information in the revised manuscript.

Moreover, it is important to emphasize, as we have done in the manuscript, that the estimation of quartz percentage is very crude and is intended to serve primarily as a 'warning flag' rather than a precise measurement. |
| RC6 | *Topographic shielding: The authors state that topographic shielding likely creates a bias towards too low erosion rates. Given that this bias can be up to 10%, it seems like a good idea to remove shielding from the erosion rate calculations once and for all. The authors argue that this is not feasible given the high computational cost. Is the re-calculation of erosion rates really so computationally expensive? As far as I understand, the computationally-expensive part is the pixel-based averaging performed on DEMs. But* |

| | |
|---|---|
| | *that part is already done. Therefore, shouldn't you be able to recalculate erosion rates fairly quickly without shielding and with time-variant scaling scheme, with the output parameters from CAIRN in a different calculator?* |
| AR6 | See our responses in AC1 and AC3.

Certainly re-calculating rates in the Balco online calculator without shielding would be quick given all the data we provide in OCTOPUS. However, we return to the question we posed in AC3. Should we do this calculation or should the users do it? |
| RC7 | *L 54: What is UOW? I don't see this defined.* |
| AR7 | UOW stands for the University of Wollongong. It is not defined here as UOW is part of the name of the data collection. |
| RC8 | *L54-48: I'm confused. What is the purpose of CRN Large Basins and DRN Denudation UOW? The article mentions that these include the published denudation rates. If that is the only reason for the existence of these two collections, why aren't the published rates added as fields to CRN International&Australia? Please, clarify.* |
| AR8 | CRN Large Basins and CRN Denudation UOW (in preparation) are two data collections that hold $^{10}$Be and $^{26}$Al data that we did not wish to include in CRN International and CRN Australia for various reasons – we explain this in detail in our first OCTOPUS description paper (Codilean et al 2018; 10.5194/essd-10-2123-2018).

To summarise: CRN Large Basins includes studies with very large basins – too large for CAIRN and also so large that one might question the meaningfulness of calculating denudation rates for these basins. Nevertheless, for completeness, we wanted to have a collection where we compile this data and let users decide how to handle them. CRN Large Basins includes the published denudation rates but no CAIRN-recalculated denudation rates.

CRN Denudation UOW (in preparation) is a repository of University of Wollongong samples that we are currently working on. |
| RC9 | *L82: Please, state the AMS standard for normalization.* |
| AR9 | We will add this information to the revised manuscript. |
| RC10 | *L83: It would be nice to have approximately 2-3 sentences telling the reader about the main characteristics of CAIRN: pixel-based production rate, exponential approximation of production rates, topographic shielding, etc.* |
| AR10 | We will add this information to the revised manuscript. |

---

## Author Comment (AC2)

RC – Reviewer comment
AR – Author response

| | |
|---|---|
| RC11 | *I agree with the authors that calculating erosion rate with a calculation method that suits the study best and has the most updated parameters is advisable. However, my assumption is that Ocotpus as a data base will be used by many scientists that might lack the detailed knowledge to choose between methods and their biases (or wouldn't do the calculation out of convienience).*

*Therefore, my suggestion would be to add the output from Balco calculators to the Octopus data table. Columns to include for ersion rates could be (1) the CAIRN-St erosion rate, (2) Balco-St without shielding, (3) Balco-Lm or LSD with shielding, and (4) Balco Lm/LSD without toposhielding.* |
| AR11 | (1) For users that lack the detailed knowledge to choose between scaling schemes etc., it is probably better to include less options, rather than more, so that we avoid even more confusion or inconsistencies on how the OCTOPUS data is used.

(2) Regarding topographic shielding, the box plot below shows percent differences between denudation rates calculated using the LSDn scaling scheme for all CRN Int and CRN AUS data with and without correcting for topographic shielding. Red lines indicate the median uncertainties (both internal and external) on the calculated denudation rates. In the case of Be-10, ~99% of the data have differences between shielding and no-shielding that are below ~6%, and below the median external uncertainty on the calculated denudation rates (~7.7%). The median difference is only ~1% and the interquartile range is 0.3 to 2.6%.

[Figure]

Given (1) and (2) above, we suggest including [*CAIRN-St*], [*Balco-LSDn with shielding*] and [*Balco-LSDn without shielding*], and **not** including [*Balco-St without shielding*]. |

| RC12 | *The authors could then have an explanatory paragraph in the manuscript stating that for studies that focus on areas with medium to high erosion rates, the St scalings are OK. When comparing global erosion rates, or when including very low erosion rates in a data set, the Lm/LSD rates are more advisable. For steep catchments with non-uniform topography and/or quartz-distribution topographic shielding should be used, whereas it should otherwise be neglected.* |
|---|---|
| AR12 | We will update the text with more information and might also include a comparison plot between St and LSDn, acknowledging that this will duplicate Greg Balco's blog post and the RIVERSAND paper (https://doi.org/10.1017/rdc.2023.74).

Regarding topographic shielding we will refer readers to DiBiase (2018; doi: 10.5194/esurf-6-923-2018) rather than provide recommendations. |

---

## Author Comment (AC3)

*Preprint gchron-2024-28*
*Short communication: Updated CRN Denudation datasets in OCTOPUS v2.3*

We wish to thank the reviewer for the detailed comments. We provide answers to each point below and, as with the first reviewer, we are happy to answer follow-up questions that may emerge from our responses.

RC – Reviewer comment
AR – Author response

| RC13 | *First, some general comments on reviewing papers about software and data sets. Basically, this should focus on whether the paper correctly describes what the software/data does/is. It's not really fair for reviewers to demand additional software features or tell the author to process the data in some totally different way. Unfortunately, this means that the most insanely maddening aspect of the OCTOPUS website -- the fact that I can't just click on a sample name displayed on the webpage and get a simple listing of the data associated with that sample -- is not in bounds for this review. Really, I think this is completely nuts -- it is bonkers to have to download a large data set, or hack the Geoserver XML responses, just to look at the data for an individual sample. And it is such a simple piece of functionality to add live links to data pages to the sample name popups that I am utterly mystified as to why this isn't done. However, none of this is allowable in a paper review, and I am obligated to stick to the basic function of evaluating whether the paper correctly describes the data set.* |
|---|---|
| AR13 | Thank you for this acknowledgement. Indeed, it is frustrating when reviewers or commentators come with a shopping list of items that they would like to see added. The OCTOPUS system architecture is complex (see below) and adding new data or making changes publicly available is more involved than simply updating a spreadsheet. |

[Figure]

| | Our long-term goal is to democratise the OCTOPUS project and so we are more than happy to invite the reviewer (and any other member of the scientific community) to join the OCTOPUS GitHub repository (https://github.com/octopus-db) and contribute with code. For security reasons the OCTOPUS code base is currently sitting in a private repository. |
|---|---|
| RC14 | *1. The relationship of 'OCTOPUS' and the various OSL, palynology, and C-14-related data sets that are briefly touched on in the introduction is unclear. It is clear that the interests of the author lie primarily in the area of erosion-rate applications of cosmogenic-nuclide data, which means that in the context of this paper the OSL and other data sets appear basically an afterthought. In addition, the title of the paper is 'updated cosmogenic-nuclide data' and not 'updated OSL data', but the discussion around line 30 indicates that there are some updates to the OSL data as well. As cosmogenic-nuclides-in-detrital-sediment data, OSL, and paleoecology data are really not very similar, and in many ways the challenges of storing OSL data in an organized way are much greater, my suggestion is for the author to just write papers about these things separately -- cover only the cosmogenic-nuclide data in this paper and then write a different paper in which the other data sets can be discussed in useful detail.* |
| AR14 | Our aim in the introduction section is to provide a brief history of OCTOPUS releases. We mention the updated OSL data and the other collections for completeness and clarity and point the reader to papers or manuscripts that describe these collections in detail:

 • SahulArch: Saktura et al. 2023 https://doi.org/10.1080/03122417.2022.2159751
 • SahulChar: Rehn et al. 2024 https://doi.org/10.5194/essd-2024-328
 • IPPD: Herbert et al. 2024 https://doi.org/10.5194/cp-2024-44

 A manuscript focusing on the OSL and TL data in SahulSed is in preparation. This *Geochronology* manuscript focuses on the updated CRN data and so we are already doing what the reviewer is recommending. |
| RC15 | *2. What are 'partner' datasets (around line 23), and why were these two specific data sets selected from the much larger universe of geochronology databases? Why not just include anything with a geodata type feed (ICE-D, Earthchem, USGS Geochron)? Is this, like, an endorsement deal? Did money change hands?* |
| AR15 | Partner collections are data that we have agreed to host on the OCTOPUS platform but have not committed to maintaining. We are more than happy to include other collections, and for example ICE-D would be a nice addition and perhaps we can have a conversation on how best to do this. USGS Geochron is an awesome resource – we were not aware of this and wish to thank the reviewer for pointing it out to us. There is a similar effort led by AuScope (https://www.auscope.org.au/ausgeochem) and we had some preliminary conversation about making AuScope Geochem and OCTOPUS work together.

 At the end of the day, however, we have finite temporal and financial resources and cannot do everything without help from the community. |
| RC16 | *3. The relationship of 'collections' to data sets is very ambiguous. In one context (line 30-ish), a data type (e.g., OSL or paleoecology) is a 'collection,' whereas later discussion (line 45, 55 areas) then reveals that a single data type is composed of multiple 'collections.' Why is this?* |

| | |
|---|---|
| | *This confusing nomenclature is inherited from the earlier versions of OCTOPUS, where it was also confusing, and it would be helpful if it was explained more clearly here.* |
| AR16 | We will make changes to the text in order to improve clarity. |
| RC17 | *The discussion around line 68 of why there are some studies that are not included in the database is mystifying, and somewhat concerning. The whole point of developing a centralized online database is that the 'return on investment' (line 69) is very high no matter how few data points there are in a study, because you only have to read the paper and ingest the data once, and then you are done. Also, of course, the computational effort scales with the number of samples, so studies with fewer data are actually less of an investment. Thus, this argument is not at all compelling; in fact, at face value it seems kind of bizarre. Furthermore, this discussion gives the idea that data sets with fewer data are deemed to be of lesser quality, which of course is a terrible approach from the science perspective. From the perspective of using the data for actual Earth science (rather than, e.g., bibliometrics) applications, how the data are grouped into 'studies' is totally arbitrary and irrelevant, so data selection should not be based on this grouping. In my view this section of the paper reveals a significant weakness of the database implementation. As this is written, it's also kind of a weird threat: include more than ten data points in your paper or else it will not be included in the database. As inclusion of data in databases of this sort is a significant element in subsequent data discoverability and reuse, this is a terrible message to send not only from the scientific perspective, but also from the perspective of outreach and student/early career development. Honestly, if I were the funding agency I would squelch this approach immediately.* |
| AR17 | There are a number of misconceptions in this comment that we unpack below:

• To clarify, the funding that we have received from the Centre of Excellence for Australian Biodiversity and Heritage, and that we acknowledge in the manuscript, was to cover migration to Google Cloud, database running costs (i.e., monthly fees for Google Cloud hosting) and the development of SahulSed, SahulArch, SahulChar and IPPD. In 2016 we obtained funding to develop the CRN collections that were released as part of OCTOPUS v.1. Since then maintaining the CRN collections has been a voluntary effort by the two authors of this manuscript with no funding available for this purpose. In this context '*return on investment*' is referring to the time we volunteer to maintaining the database.

• The computational effort does not always scale with the number of samples. In fact, for the reasons described next, papers with fewer data often need more effort to ingest. Recalculating basin-wide denudation rates involves a lot of detective work to do with identifying and delineating the basins from where samples were collected. This works best when a publication (1) includes sample coordinates with sufficient precision, a detailed map identifying each sampled basin, and information such as basin areas and/or sample site elevations, and (2) the different pieces of information match. In our experience, most studies (although there are exceptions) with small number of CRN data points (n ~ 3 or 4) do not allocate a lot of space to documenting this data in sufficient detail for reproducibility. We do our best to ingest all available data, and contact the corresponding authors for help, but this is often futile. As we mention in our manuscript, we have identified about 47 studies that we may never be able to incorporate due to lack of information. |

| | |
|---|---|
| | • Grouping data into studies offers a practical way of organising the large amounts of raster data and also the CAIRN input/output files that we include in the database. The actual tabular data stored in the relational database is, of course, seamless and the link to studies is achieved using a unique ID (STUDYID). The raster layers come in various spatial resolutions and are projected to different UTM zones depending on the size and location of each study area. Therefore grouping data into studies allows for these raster layers to be containerised and served for download. We discuss this point in previous papers describing OCTOPUS. |
| RC18 | *5. The discussion about how data sets from cratonic areas are sparse should probably make note of the fact that river systems in these areas are largely depositional rather than erosional systems, in which case cosmogenic-nuclide erosion rate measurements actually don't work. Really what is wanted here is an assessment of how representative these measurements are of the area in which the measurements could be made, not the area in which the measurements could not be made.* |
| AR18 | We see the point of the reviewer here. However, all we are trying to do in the offending sentence (i.e., "*However, data from low-gradient, tectonically passive regions remain sparse, particularly in Africa*") is to point out the lack of data from the African continent. |
| RC19 | *6. The discussion in line 105-ish should be more clear about the fact that the mean atmospheric pressure is not derived from an atmosphere model (as one might reasonably expect) but by inverting the scaling model. That is, one should not assume that the mean atmospheric pressure in this field has any meteorological significance. It sort of says this, but this point should be made clear.* |
| AR19 | This is a good point. We will change the last sentence of the paragraph starting at line 105 to make this point more clear. |
| RC20 | *7. Near line 118, later on this page, and in Figure 3, I don't understand why calculations are being made at the location of the basin outlet. If you are able to calculate the mean elevation, then you obviously know where the basin is, and can also calculate the centroid latitude. In what circumstance would you ever care about the location of the basin outlet, or want to use that in a calculation?* |
| AR20 | This is also a good point. Our aim here was to see how bad things can get – but we acknowledge that this is probably an unlikely scenario and most (if not all) users will calculate basin centroid coordinates and use those. |
| RC21 | *8. Line 120-123 is slightly misleading. These differences are not because the calculation methods (Balco 2008 vs. CAIRN) are different, they're because the mean basin elevation is different from the effective elevation. This should be clarified.* |
| AR21 | Good point, again. We will clarify this in the text. |
| RC22 | *9. The section at the bottom of p. 6 (lines 135-140 area) is a little bit incoherent, because it is not clear what each comparison is supposed to test. It would be helpful to rewrite this to make clear what assumption is being tested with each comparison, e.g., something like, 'To quantify the effect of using the mean elevation vs. the effective elevation, we did X and here* |

| | |
|---|---|
| | *are the results. To quantify the differences between CAIRN and Balco 2008 with the same input parameters, we did Y and here are the results.' You get the idea.* |
| AR22 | Indeed, the last paragraph on page 6 is somewhat incoherent. We will change the text to better explain what we are trying to achieve here. |
| RC23 | *10. I agree with the other review that the paper should make clear that the glacier cover fraction and quartz-occurrence-inferred-from-lithology fields are suitable for general guidance, but probably not very quantitative.* |
| AR23 | We feel that we are already doing an adequate job here.

We mention on lines 153-155 and 160-162 that our GLIMS corrected rates are end-member maximum values and are meant for providing an insight "*into the potential influence of glaciers on the overestimation of $^{10}$Be-derived denudation rates*".

Regarding quartz occurrence we provide ample caution including this sentence at lines 181-183: "*To reiterate, for most basins, these data are too coarse to enable precise corrections to the recalculated denudation rates; however, they may still offer valuable insights and help identify basins where such corrections could be warranted.*" |
| RC24 | *11. The y-axis in Fig 5 is labeled such that it looks like GLIMS is being subtracted from CAIRN, which doesn't make any sense. Also, if we assume that it is really two erosion rates that are being subtracted, then the units should be m/Ma, not percent. This needs to be labeled in a way that conforms with the units (something like 100 * (E_glaciers - E_noglaciers)/E_noglaciers ?).* |
| AR24 | The y-axes in Figs. 3 and 5 and both x- and y-axes in Fig. 4 represent percent difference between two denudation rate values, calculated as: [(D1-D2)/mean(D1,D2)] x 100.

In the case of Fig. 5, spelling this out would make the label:

[(E_glaciers – E_nonglaciers) / mean(E_glaciers, E_nonglaciers)] * 100

The above would be too long to fit. As a compromise we went with CAIRN – GLIMS [%] and we explain in the figure caption what this actually means. We could substitute with 'Percent difference [%]' in all figures but then we lose information on what the sign (negative or positive) means. |
| RC25 | *12. Again following discussion of quartz-fraction estimate in other review, the discussion in line 180-ish is not helpful without some idea of which lithologies are and are not quartz-bearing. Perhaps the easiest way to handle this would be just to have a supplemental table indicating which GLiM classifications are and are not considered to have quartz.* |
| AR25 | Good suggestion. We will include a table indicating the GLiM classification. |
| RC26 | *13. In line 190-ish, the discussion here basically says that the topographic shielding calculations are wrong, but we did them anyway, which is rather odd. Again, the computational-resources argument is not super compelling, but on the other hand this* |

*paragraph does appear to correctly describe what has been done, so even if what was done is questionable, this section passes the test of whether it is accurately described.*

| AR26 | Strictly speaking (1) calculating topographic shielding by not accounting for the change in shielding with depth – what is done in OCTOPUS, and (2) avoiding topographic shielding corrections altogether, including in steep basins where quartz distribution and / or denudation rates are not uniform, are both incorrect.

What we are saying in AR11 is that (1) the effect of calculating or ignoring shielding is trivial (~99% of the data have differences between shielding and no-shielding that are below ~6%, and below the median external uncertainty on the calculated denudation rates), and (2) this does not warrant spending two months on recalculating everything with CAIRN.

Furthermore, users are provided with the means of doing the recalculations themselves if they wish. |
|---|---|
| RC27 | *14. Around line 205, there needs to be more explanation of whether basins with internal drainage issues are (i) not recorded at all in the database (which in my view is bad practice) or (ii) recorded in the database with Be-10 concentrations, etc., but with no associated calculated erosion rate (preferable). Also, frankly, I don't really understand what the problem is here, because Figure 6 makes it clear that you do know what the actual correct drainage basin boundary is in lat/long coordinates - why is it possible to project the black lines into UTM but somehow impossible to project the blue lines? Also, of course, for pixel-based production rate calculations, you don't actually have to project out of lat/long - you can just weight by the actual area of each cell in real units. But that would probably require a redesign of the whole thing.* |
| AR27 | We only exclude basins (and data) where we are unable to reproduce with confidence the drainage basin as described in the source publication, and we have convinced ourselves (for example based on satellite imagery, etc) that our inability to reproduce the basin is not due to mistakes in the source publication. The number of such basins is small (n < 20).

The issue described in the final paragraph of Section 5 (lines 197 – 205) describes a limitation of CAIRN:  it delineates the basins from a DEM that needs to be projects in one of the WGS84 UTM zones. One cannot provide basin outlines as a vector file to aid in basin identification (although one could clip a DEM using basin outlines and this provides a solution in some but not all cases). All that CAIRN takes as input is the DEM and the sample locations. It then delineates basins etc by snapping the sample locations to the nearest channels – as defined using stream order and upstream contributing area thresholds.

Fig. 6 shows drainage basins **as derived** from two different DEMS (one hydrologically enforced and the other after the former was projected to UTM). The basin boundaries are not the ones being projected as the reviewer seems to believe in RC27. Following reprojection of the DEM, CAIRN is using its own sink filling and flow routing algorithm to derive a hydrologically connected drainage net that results in internally drained areas being connected to the main drainage. Using the blue lines in Fig.6 to clip the DEM and remove internally drained cells prior to reprojecting may result in CAIRN not being able to produce a drainage net that connects all upstream areas to the basin outlet. |

---

## Author Comment (AC4)

*Preprint gchron-2024-28*
*Short communication: Updated CRN Denudation datasets in OCTOPUS v2.3*

We thank the reviewer for taking the time to thoroughly read and evaluate our manuscript. Several points raised in this review align with comments made by the other two reviewers. In such instances, we refer to our detailed responses provided in AC1, AC2, and AC3.

RC – Reviewer comment
AR – Author response

| | |
|---|---|
| RC28 | *I myself still find the usability a bit frustrating, but admittedly I haven't spent much time trying to learn the ins and outs.* |
| AR28 | As noted in AR13, we extend an invitation to the reviewer to join the OCTOPUS GitHub repository and to channel any frustrations into constructive contributions to the OCTOPUS project, thereby helping to enhance the user experience. |
| RC29 | *Line 17-43: The introduction describes a number of features of the OCTOPUS database that are not relevant to the current paper – (luminescence, charcoal, 10Be and 26Al exposure ages, pollen, etc.). This material should be shortened and the introduction centered on catchment CRN which is the focus of this paper.* |
| AR29 | As highlighted in AR14, the introduction aims to provide a concise history of OCTOPUS releases. This places the current updates into context and appropriately acknowledges previous publications that discuss aspects of the database in detail, which are not addressed in this manuscript. |
| RC30 | *Line 68: "limited return on investment"? I would not dismiss these studies from a database just based on the number of samples. Some may be in places with limited data coverage and thus quite valuable. I understand the resource limitations, but since these studies have already been identified I would recommend including on the OCTOPUS website a list of all identified studies, with an indication of their status (included, in process, not in process). This might streamline the possibility of a user to request addition of an important, but small study to the database?* |
| AR30 | As discussed in detail in AR17, we do not automatically dismiss any study solely based on it including a small number of data points. However, we assign lower priority to studies with limited data points and insufficient information to facilitate the straightforward recalculation of denudation rates.

We recognize the value of maintaining a public record of all identified studies and will explore ways to achieve this while ensuring that authors feel included. |
| RC31 | *Line 110: A quick note about how atmospheric pressure is actually derived would be helpful (I think CAIRN is starting with elevation, mapping to atmospheric pressure using NCEP2 reanalysis, and then back-calculating and effective pressure from Stone 2000, right?)* |
| AR31 | Good suggestion. We will add more information to the revised manuscript regarding how atmospheric pressure is calculated in CAIRN. |

| | |
|---|---|
| RC32 | *Line 119: It's perhaps confusing to compare between "mean elevation" and the "effective atmospheric pressure", since the issue is not one of using elevation or pressure, but going through the process to calculate the "effective" pressure (or elevation).* |
| AR32 | In the absence of being provided with effective pressure or effective elevation values for each basin, most users would probably default to using the mean basin elevation – which was always included in the OCTOPUS database – rather than going through the effort of using the included DEMs to calculate effective pressure / elevation values.

Furthermore, the effective atmospheric pressure values included in the updated CRN Denudation datasets are approximations of the pixel-by-pixel calculations performed by CAIRN. Consequently, Figures 3(a) and 3(b) compare one approximation (effective pressure values) with another approximation (mean elevation) when calculating denudation rates, then evaluate both against CAIRN's more rigorous pixel-by-pixel approach.

In our view it is important to show that while not perfect, effective pressure performs better than mean basin elevation. It is also important to illustrate at which point mean basin elevation ceases to be a suitable proxy for calculating basin-wide denudation rates. |
| RC33 | *Line 129 and Figure 4: It seems unnecessary to discuss using outlet latitude, since the centroid latitude is already available and clearly the better approximation for production rates. And Figure 4 seems unnecessary as well.* |
| AR33 | As we mention in AR20, we agree that centroid latitude is the better approximation and because it is easy to calculate, most people will chose this over outlet latitude. However, we disagree with the reviewer that Figure 4 is unnecessary.

There may be instances where both mean elevation and sampling latitude are available (for example provided in the original publication) but it is not possible to delineate the actual drainage basin with confidence, and thus not possible to calculate centroid latitude. The latter could be due to insufficient information provided or issues with the DEM available (e.g., delineated basin does not match published basin). Figure 4 is useful in this regard as it shows that using centroid latitude vs. outlet latitude only becomes important when basin areas start exceeding $10^4$-$10^5$ km$^2$, and so it may be possible to calculate reliable denudation rates using mean elevation and sampling latitude if certain basin relief and basin area criteria are met. |
| RC34 | *Figure 3: The axis labels are hard to interpret without the caption – It would help readability to add some plain language to the labels ("Calculated erosion rate difference", "Erosion rate", etc.)* |
| AR34 | See our response in AR24 regarding axis labels. We acknowledge that, without the figure caption, it is currently challenging to interpret the meaning of the axis labels in Fig. 3. We will explore the best approach to modifying the labels to enhance clarity. |
| RC35 | *Line 150-167: There are so many issues with interpreting CRN data from currently glaciated basins (non-uniform erosion rates, time-varying erosion, storage/reworking in moraines, etc.) I would personally avoid doing this calculation at all and just flagging catchments that contain glaciers.* |

| | |
|---|---|
| AR35 | The presence or absence of glaciers is less critical than the areal extent of the ice. Therefore, merely flagging basins that currently contain glaciers, without providing additional details, may not be particularly informative. Factors such as sediment storage and reworking in moraines are equally, if not more, relevant for basins that were glaciated in the past but are now ice-free. Therefore, the issue of moraines is more complicated and goes beyond present day ice coverage, and is one that we cannot easily address at the global scale.

Fundamentally, however, the impact of glaciers on cosmogenic nuclide concentrations exported from a basin is one of dilution. Glaciers shield parts of the basin from cosmic rays and contribute material with depleted cosmogenic nuclide concentrations, thereby diluting the overall signal. The degree of this dilution depends on numerous factors and may be impossible to estimate with high confidence. However, it is possible to estimate a worst-case-scenario and this is what were are including in the updated datasets.

We explicitly acknowledge the limitations of our calculations in the text and emphasize to readers that the primary purpose of this new data is to facilitate the evaluation of data quality, rather than to be used as definitive values. |
| RC36 | *Line 180: what is the definition of "quartz-bearing"?* |
| AR36 | See our responses in AR5 and AR25. We will add more information in the revised manuscript regarding GLiM lithology classes used to identify quartz-bearing rocks. |
| RC37 | *Line 187-196: I agree that the topographic shielding corrections are going to be minimal for most catchments, but it is nonetheless quite awkward to be baking in an erroneous correction into every denudation rate. It can't be \*that\* hard to fix this? Or at least acknowledge that it needs to be fixed in the future?* |
| AR37 | See our responses in AR6, AR11, AR12, and AR26.  In AR11, we propose including, alongside the CAIRN-calculated denudation rates, those calculated using the Balco calculators both with and without topographic shielding corrections. In doing so, we are addressing this issue. |